# Interstitial Macrophages Lead Early Stages of Bleomycin-Induced Lung Fibrosis and Induce Fibroblasts Activation

**DOI:** 10.3390/cells12030402

**Published:** 2023-01-24

**Authors:** Sofia Libório-Ramos, Catarina Barbosa-Matos, Raquel Fernandes, Caroline Borges-Pereira, Sandra Costa

**Affiliations:** 1Life and Health Sciences Research Institute (ICVS), School of Medicine, Campus Gualtar, University of Minho, 4710-057 Braga, Portugal; 2ICVS/3B’s—PT Government Associate Laboratory, 4806-909 Braga, Portugal

**Keywords:** pulmonary fibrosis, interstitial macrophages, alveolar macrophages, conditioned medium, fibroblasts, myofibroblasts

## Abstract

A progressive fibrosing phenotype is critical in several lung diseases. It is irreversible and associated with early patient mortality. Growing evidence has revealed pulmonary macrophages’ role as modulators of the fibrotic processes. The proportion, phenotype, and function of alveolar (AM) and interstitial macrophages (IM) at the early stages of bleomycin-induced pulmonary fibrosis have not been clearly described. In this way, our study aimed to characterize these macrophage populations and investigate the effect on fibroblast activation. C57BL/6 mice were intratracheally injected with bleomycin and were sacrificed at day 3, 5, and 7 for the performance of flow cytometry and fluorescent-activated cell sorting analysis for protein and gene expression quantification. After bleomycin administration, the proportion of IM was significantly higher than that of AM, which showed a decay during the inflammatory phase, and peaked at day 7. At day 7 of the inflammatory phase, AM started shifting their phenotype from M1-like towards M2, while IM showed a M2-like phenotype. Conditioned medium derived from IM sorted at day 7 induced fibroblast activation and differentiation in myofibroblasts in vitro. Our findings indicate that IM are the largest macrophage population at the early stages of experimental pulmonary fibrosis and are secreted mediators able to activate fibroblasts, pointing to macrophage modulation as a potential therapeutic strategy to restrain progressive fibrosing lung disorders.

## 1. Introduction

Idiopathic pulmonary fibrosis is the most severe and fatal form of fibrosing lung diseases, with a median survival term of only 2 to 3 years after diagnosis [1,2]. Even with the poor prognosis of this disease, its etiology remains undetermined. Despite the poorly defined pathophysiology, it is thought to be caused by a chronic injury in the alveolar epithelial cells induced by aberrant wound healing [3]. Unfortunately, current therapeutic strategies remain limited and can delay the disease progression rather than provide a cure [4]. Therefore, there is an urgent need to develop new and efficient therapeutic strategies.

In addition to occurring in idiopathic pulmonary fibrosis, pulmonary fibrosis is also a common denominator of several lung disorders and is characterized by the accumulation of myofibroblasts and excessive deposition of extracellular matrix [1,5,6]. These processes culminate in the formation of a fibrotic ‘scar’ which leads to the irreversible destruction of lung architecture and, ultimately, to death due to respiratory failure [1,5,6]. There is increasing evidence that the immune system is involved in the progression of this pathologic condition. Macrophages are particularly prominent at sites of inflammation and fibrosis, rising as a crucial cell population in these processes [7,8].

During adulthood, macrophages are the central innate immune cells in the lung under homeostatic conditions and are known as the first line of defense in many organs [9]. Diverse and distinct macrophage populations exist in the lung. Resident alveolar macrophages (AM) are mainly derived from precursors in the embryo and populate the airway space soon after birth [10,11,12]. In a steady state, this population is capable of self-renewal and persists in the lung, the most represented macrophage population in normal adult mice lungs [13]. Moreover, a population of recruited AM arises postnatally from circulating bone marrow-derived monocytes [14]. AM are the best studied and understood lung macrophage population. However, interstitial macrophages (IM), infiltrated within the lung parenchyma, are rising as critical players in adult and murine lung fibrosis [7,13,15]. The origin and function of IM are still not completely clarified, partly due to the difficulties in isolating this population [13,16]. It is believed that IM originate both from an embryonic yolk sac origin and postnatal bone marrow origin [10,17].

There is increasing evidence of the importance of interstitial and alveolar macrophages to pulmonary fibrosis [18,19,20,21]. However, mainly due to their heterogeneity and plasticity, depending on the microenvironment, which confers them the capacity to exert tissue-specific functions according to the transitory needs, but also due to, for example, the M1/M2 dichotomic metabolic state, among others (7, 8), their role may be controversial. Furthermore, macrophages can switch between two different activation states, which dictate their modulation of fibroproliferative responses. It has been suggested that classically activated macrophages (M1) maintain tissue inflammation, while alternatively activated macrophages (M2) play a role in resolving lung inflammation and aberrant wound healing cascade during fibrosis [9,22,23].

Previous studies suggested that macrophages are present and adapt their phenotype and function when fibrogenesis is dominant. However, the detailed individual contribution of macrophages, both from the alveolar and interstitial compartments, has not been fully evinced in the early stages of the disease progression. Here, we use a bleomycin (BLM)-induced pulmonary fibrosis mouse model to characterize alveolar and interstitial macrophages in the initial stages of disease progression. In addition, we provide in vitro data to determine the paracrine effect played by IM in the advancement of lung fibrosis.

## 2. Materials and Methods

### 2.1. Mice

Eight-week-old gender- and age-matched C57BL/6 mice were bred under specific pathogen-free conditions and kept at the Life and Health Sciences Research Institute (ICVS) Animal Facility. Mice were fed ad libitum and kept under light/dark cycles of 12 h, temperature of 18–25 °C, and humidity of 40–60%.

All procedures in vivo followed the EU-adopted regulations (Directive 2010/63/EU). The ethical and regulatory approvals were consented to by the Direção Geral de Alimentação e Veterinária (DGAV, ref. 003671) and conducted according to the guidelines sanctioned by the Portuguese ethics committee for animal experimentation, DGAV.

### 2.2. Bleomycin-Induced Pulmonary Fibrosis Mouse Model

Male C57BL/6J mice, 10 to 12 weeks old, were anesthetized with an intraperitoneal injection of ketamine and medetomidine, followed by intratracheal administration of BLM sulphate (BML-AP302-0010, Enzo Life Sciences, Miraflores, Portugal) as a single dose of 2 mg/kg body weight dissolved in 50 µL saline (treated) or, as control, with saline alone (untreated)—day 0. Weight was measured every single day throughout the experiment. At 3, 5, and 7 days after BLM administration, mice were euthanized. Mice were sacrificed by excessive doses of anesthesia to collect bronchoalveolar lavage fluid (BALF) and single-cell suspensions. Regarding the number of animals, 7–10 mice (per timepoint) were injected and sacrificed at day 3, 5, and 7 for flow cytometry analysis. For fluorescence-activated cell sorting, performed at day 7, 15 mice were injected with saline (untreated), and 15 mice were injected with bleomycin.

### 2.3. Bronchoalveolar Lavage (BAL)

In brief, the trachea was exposed by a midline incision and cannulated. Bilateral BALF was recovered by lavage three times with 1 mL PBS 1×. After collection, BALF cells were maintained on ice to be then added to single-cell suspensions obtained from tissue dissociation.

### 2.4. Preparation of Lung Single-Cell Suspensions

Briefly, mice were over-anesthetized intraperitoneally and perfused with 20 mL of sterile PBS 1×. After dissection, lungs were placed in dispase (11553550, Corning, Corning, NY, USA) while being minced into 2–3 mm^3^ pieces for 20 min at room temperature. Then, they were also digested in Dulbecco’s Modified Eagle Medium (DMEM) 1× (21969-035, Gibco, Waltham, MA, USA) containing 1 mg/mL DNase I (10104159001, Roche, Basel, Switzerland) and 5 mg/mL collagenase II (17101-015, Gibco). After an enzymatic digestion period of 10 min at 37 °C, tissue disruption was performed by mechanical titrations with P1000 tip, and then a second incubation period of 10 min was performed at 37 °C. After incubation, the disrupted tissue suspension was collected and sequentially filtered through 100 µm and 70 µm cell strainers. The pellet was resuspended in ACK lysis buffer (A1049201, Gibco), which was then incubated for 4 min. Cells were then resuspended in cold PBS 1× containing 4% fetal bovine serum (FBS-10270-106, Gibco). These lung single-cell suspensions were used to analyze the expression of various surface markers for immune cells by flow cytometry and magnetic (MACS)- and fluorescence (FACS)-activated cell-sorting-based separation.

### 2.5. Flow Cytometry

In brief, lung cells were harvested, washed, and resuspended in cold PBS containing 4% FBS. Then, 1 × 10^6^ cells were preincubated with TruStain FcX anti-mouse CD16/32 antibody (Biolegend (San Diego, CA, USA), 101320) for 10 min at room temperature. This blocking of non-specific binding was followed by incubation for 30 min on ice, in the dark, with the following anti-mouse antibodies: BV510-conjugated CD45 (1:100—103138, Biolegend), PE-conjugated SiglecF (1:200—552126, BD Biosciences, San Jose, CA, USA), PE/Cy7-conjugated CD11b (1:100—101216, Biolegend), FITC-conjugated CD64 (1:200—139316, Biolegend), PerCP/Cy5.5-conjugated CD11c (1:100—117328, Biolegend), BV605-conjugated Ly6G (1:100—127639, Biolegend), Alexa Fluor 647-conjugated I-Ab (1:100—115310, Biolegend), APC/Cy7-conjugated CD24 (1:100—47-0242-80, Invitrogen, Waltham, MA, USA), BV650-conjugated CD80 (1:100—104731, Biolegend), and BV786-conjugated CD206 (1:100—141729, Biolegend). Then, cells were incubated with Viable Dye (VD) eFluor^®^ 450 (65-0863-14, EBioscience, San Diego, CA, USA) for 30 min at 4 °C in the dark. Cells were resuspended and fixed in 4% paraformaldehyde for 20 min on ice in the dark.

Regarding intracellular staining with PE-Cy7-conjugated TNF-α (1:100—506323, Biolegend), before the membrane staining, cells were preincubated for 4 h at 37 °C with 10 ng of Phorbol 12-myristate 13-acetate (P8139, Sigma-Aldrich, St. Louis, MO, USA), 100 ng of ionomycin (I0634, Sigma-Aldrich), and 2 µg of brefeldin A (B7651, Sigma-Aldrich) in DMEM 1× supplemented with 10% FBS and 1% penicillin/streptomycin (P4333, Sigma-Aldrich). Then, the staining proceeded as mentioned in the previous paragraph, except for CD11b; in this case, we used APC-conjugated antibodies (1:200—553312, BD). After fixation, cells were permeabilized in 0.5% saponin in PBS 1×, and then intracellular staining was performed.

Data were acquired on a LSRII flow cytometer and analyzed using FlowJo Software. 

For FACS-based separation of IM and AM contained in the CD45-positive fraction, the following antibodies were used: anti-mouse PE/Cy7-conjugated CD45 (1:200—552848, BD Biosciences), PE-conjugated SiglecF (1:200—552126, BD Biosciences), APC-conjugated CD11b (1:400—553312, BD Biosciences), FITC-conjugated CD64 (1:300—139316, Biolegend), PerCP/Cy5.5-conjugated CD11c (1:200—560584, BD Biosciences), and APC/Cy7-conjugated Ly6G (1:300—560600, BD Biosciences).

### 2.6. Reverse Transcription Quantitative PCR 

At day 7 after BLM administration, FACS-sorted IM and AM were stored in RLT buffer and frozen at −80 °C. Total RNA was isolated from sorted interstitial and alveolar macrophages using RNeasy Plus Micro Kit following the instructions provided by the manufacturer (74034, Qiagen, Hilden, Germany).

First-strand cDNA was synthesized using the Super Script III First-Strand Synthesis Supermix kit, which was used to convert all the extracted RNA following the instructions provided by the manufacturer (18080400, Invitrogen).

Reverse transcription quantitative PCR (RT-qPCR) was conducted using the 7500 Fast Real-Time PCR System (Applied Biosystems, Waltham, MA, USA, Life Technologies, Carlsbad, CA, USA). RT-qPCR was performed using Taqman Universal PCR MasterMix (4304437, Thermo Fisher Scientific, Waltham, MA, USA) and one TaqMan^®^ Gene Expression assay.

The following probes were used for each target gene: *Gapdh* (Mm99999915_g1), *IL-1β* (Mm00434228_m1), *Tnf-α* (Mm00443258_m1), *Ccl2* (Mm00441242_m1), *Cxcl2* (Mm00436450_m1), *Ccl5* (Mm01302427_m1), *Ccl6* (Mm01302419_m1), *Ccl9* (Mm00441260_m1), *Arg1* (Mm00475988_m1), *Tgf-β1* (Mm01178820_m1), *Tgf-bi* (Mm01337605_m1), *Chi3l3/ Ym1* (Mm00657889_m1), *Ccl24* (Mm00444701_m1), *Fibronectin 1* (Mm01256744_m1), *Mmp12* (Mm00500554_m1).

Relative expression levels for each target gene were calculated using the 2^−ΔΔCt^ method; Ct values of target genes were normalized to the reference gene *Gapdh*. Data were presented in fold change, which defined the gene’s relative expression in the respective vehicle group to 1.

### 2.7. Interstitial Macrophages Fluorescence-Activated Cell Sorting and Culture

Untreated and treated animals were used to obtain interstitial macrophages for primary culture and to collect conditioned medium (CM).

The lung single-cell suspensions were incubated with anti-CD45 MicroBeads (130-052-301, Miltenyi Biotec, Bergisch Gladbach, Germany) to enrich the populations of interest, which are minor in the lung.

CD45^+^ cells were collected over MS column separation (130-042-201, Miltenyi Biotec) in the OctoMACS separator (130-042-109, Miltenyi Biotec) magnetic field. Viable cells were counted using trypan blue, and flow cytometry staining was performed to proceed further to the fluorescence-activated cell sorting procedure using BD FACSARIA II cell sorter (BD Bioscience).

Following cell sorting, 2.5 × 10^5^ macrophages in DMEM 1× supplemented with 10% FBS, and 1% penicillin/streptomycin were plated in Sterilin^TM^ Square Petri Dishes (Thermo Fisher Scientific) for 24 h at 37 °C in a humidified atmosphere containing 5% CO_2_. After this time, the CM was collected, centrifuged at 1500 rpm for 5 min, and then stored at −80 °C.

### 2.8. Culture and Stimulation of MLg Cell Line

MLg (MLg 2908, ATCC^®^ CCL206™), a mouse lung fibroblast cell line, was used in in vitro assays. MLg cells were maintained in Minimum Essential Medium (MEM; 31095-029, Gibco) supplemented with 10% FBS, 10 mM HEPES (15630-056, Gibco), and 1% penicillin–streptomycin solution. Cells were maintained at 37 °C in a humidified atmosphere containing 5% CO2. When cells reached 80–90% confluence, they were enzymatically dissociated using TrypLE Express Enzyme (12605-028, Gibco) for 5 min at 37 °C, and the cell suspension was passed to a new flask in a low dilution.

Based on Meziani et al. [19], 5 × 10^3^ or 1.5 × 10^5^ MLg cells in DMEM 1× containing 10% FBS, 10 mM HEPES, and 1% penicillin–streptomycin were seeded in a regular 96-well plate (MTS assay), µ-Plates 96 Well Black (immunofluorescence—89626, Ibidi), or in a regular 6-well plate (Western blot), respectively, and were allowed to adhere overnight. Next, these cells were cultured for 48 h with untreated and treated, sorted, IM-derived CM with 10% FBS. In control conditions, MLg cells were cultured in the same medium without contact with IM-derived CM. The supplementation with 10% FBS was necessary as MLg cells do not survive when cultured without it, not even in lower percentages.

### 2.9. Metabolic Activity Assay

MLg proliferation after stimulation with control or CM for 48 h was assessed by MTS assay according to the manufacturer’s instructions (CellTiter 96 Aqueous One Solution, PROMEGA, Milan, Italy). Briefly, MLg were incubated in control medium with 1:10 MTS for 2.5 h at 37 °C. Next, 490 nm absorbances were read using a 96-well plate reader (Varioskan Flash—Thermo Fisher Scientific, Vantaa, Finland). All assays were performed in triplicate.

### 2.10. Immunofluorescence

The proliferative and transdifferentiating state of MLg cells after CM stimulation was assessed by immunofluorescence. Following 48 h of stimulation with IM-derived conditioned medium, the medium was carefully removed, and cells were washed with PBS 1×. Then, adherent cells were fixed with PFA 4% (proliferation protocol) or methanol 100% (transdifferentiation protocol) for 30 min. Fixed cells were permeabilized in Triton-X-100 0.3% in PBS 1× for 20 min in the case of the proliferation protocol. Cells were washed twice with PBS Tween 0.5% for 10 min, and blockage of non-specific binding was performed with newborn calf serum (proliferation protocol; 1/10—186427, Biochrom) or normal goat serum (transdifferentiation protocol; 1/10—G9023-0010, Sigma-Aldrich) for 1 h. Next, cells were incubated with a rabbit anti-mouse Ki-67 (1:250 in blocking solution—AB9260, Millipore), rabbit polyclonal anti-mouse vimentin (1:150 in blocking solution—#5741, CellSignalling Technology, Danvers, MA, USA), and mouse anti-mouse α-smooth muscle actin (SMA, 1:100 in blocking solution—ab32575, Abcam, Cambridge, UK) primary antibodies overnight. Cells were washed and incubated for 2 h with goat anti-rabbit conjugated to Alexa Fluor 568 (1:250 in blocking solution—a11011, Thermo Fisher Scientific) and goat anti-mouse conjugated to Alexa Fluor 488 (1:250 in blocking solution—a11001, Thermo Fisher Scientific) secondary antibodies. Finally, cells were washed and incubated with DAPI for 5 min, mounted in Ibidi Mounting Medium (50001, Ibidi, Gräfelfing, Germany), and images were obtained with an inverted fluorescence microscope (Olympus widefield inverted microscope IX53, Olympus, Japan) using the 10× objective. Data were presented as positive cells per DAPI-positive cells.

### 2.11. Protein Extraction

The transdifferentiating state of MLg cells after CM stimulation was assessed by Western blot. Following 48 h of stimulation with IM-derived conditioned medium, the medium was carefully removed, and cells were washed with PBS 1×. Then, protein was extracted by scraping the cells after adding lysis buffer (50 mM Tris-Base, 150 mM NaCl, 5 mM EDTA, 1% Triton X-100, and 1:7 of cOmplete protease inhibitors cocktail (11697498001, Roche)). Lysed cells were collected, kept on ice for 15 min, and centrifuged at 10,000 rpm at 4 °C for 15 min. The supernatant was collected, and protein was quantified using Bradford reagent (B6916, Sigma-Aldrich). For quantification, 2 µL of protein extracts, as well as a BSA standard curve, was added to each well of a 96-well plate, followed by 98 µL of PBS 1× and 200 µL of Bradford reagent. After a 10 min incubation period, absorbance was measured at 590 nm in the microplate reader (Varioskan Flash-Thermo FisherScientific, Vantaa, Finland).

### 2.12. Western Blot Analysis

Aliquots of 20 µg of protein from MLg cells were prepared in the proportion of 1:1 with loading buffer (950 μL of Laemmli Sample Buffer 2× (161-0737, Biorad) and 50 μL 2-mercaptoethanol (63689, Millipore)) and denatured over 5 min at 98 °C. Samples were then loaded onto a 10% sodium dodecyl sulphate–polyacrylamide (SDS-PAGE) gel and transferred to Amersham Protran nitrocellulose membranes (GE10600008, Sigma-Aldrich) using the Trans-Blot Turbo Transfer System (BioRad Laboratories, München, Germany). After, membranes were blocked for 1 h with TBS 0.1% Tween containing 5% of BSA. Membranes were incubated overnight at 4 °C with the primary antibodies for vimentin (1:1000, rabbit—#5741, CellSignalling Technology), α-SMA (1:500, mouse—ab32575, Abcam), and Gapdh (1:2500, rabbit—ab9485, Abcam). After incubation, membranes were washed three times in TBS 0.1% Tween and then incubated with a secondary antibody coupled to horseradish peroxidase (HRP, 1:2000, anti-rabbit—7074P2, CellSignalling Technology and 1:3000, anti-mouse—sc-516102, Santa Cruz Biotechnology) diluted in 5% milk for 1 h at RT. Membranes were washed three times with TBS 0.1% Tween, and signal was detected in Sapphire Biomolecular Imager (Azure Biosystems, Dublin, CA, USA) using Western Bright Sirius HRP Substrate Kit (K-12043-D10, Advansta, San Jose, CA, USA). The bands were quantified in ImageJ software, and quantifications were normalized to Gapdh levels.

### 2.13. Statistical Analysis

Data were analyzed and plotted using GraphPad Prism 6.00 (GraphPad Software, San Diego, CA). All results were expressed as group mean ± standard error of the mean (SEM). A p-value of less than 0.05 was considered a statistically significant difference. For all comparisons, normality was assessed with Shapiro–Wilk test; if normality was not verified, non-parametric tests were used. Comparisons between two groups were performed either with a parametric, unpaired, two-tailed Student’s *t*-test or a non-parametric, two-tailed Mann–Whitney test. Comparisons between three groups were performed either with parametric one-way ANOVA or a non-parametric Kruskall–Wallis test.

## 3. Results

### 3.1. Inflammatory Phase of Bleomycin-Induced Pulmonary Fibrosis Is Characterized by a Predominance of Interstitial Macrophages

Intratracheal injection of BLM or saline was performed on day 0. BLM administration was followed by an inflammatory phase with a peak at day 7. So, to elucidate the potential involvement of IM and AM throughout the inflammatory phase of the BLM-induced pulmonary fibrosis model, lung macrophage populations at days 3, 5, and 7 were analyzed by flow cytometry (Figure 1A). The gating strategy used (Figure 1B), based on Misharin et al. [20], is shown in Figure 1B. First, the debris and doublets were excluded using forward vs. side scatter area (FSC-A vs SSC-A) and a FSC-A vs. FSC height plot, respectively. Then, live immune cells were identified by negative staining for viable dye and positive staining for CD45. A sequential gating strategy was first used to exclude neutrophil (CD11b^+^Ly6G^+^) and eosinophil (CD11c^−^SiglecF^+^) populations, followed by the identification of alveolar macrophages (MHCII^+^CD64^+^SiglecF⁺) and interstitial macrophages (MHCII^+^CD64^+^ SiglecF^−^).

The macrophage populations’ abundance is illustrated in Figure 1C. The percentage of AM (CD64^+^SiglecF⁺) was reduced in BLM-treated lungs compared with in the saline lungs at days 3, 5, and 7 (Figure 1C). On the other hand, the frequency of IM (CD64^+^SiglecF^-^) was higher in BLM-treated mice than in the untreated mice (Figure 1C) at all time points, being highest at day 7.

Moreover, we observed that the most prevalent macrophage population in BLM-treated animals was that of IM (Figure 1D). Their percentage duplicated over the days after BLM administration from 7.2 ± 1.9% on day 3 to 13.8 ± 1.5% on day 7. On the contrary, no differences were observed in the population of AM. In the BLM-treated animals, the proportion of IM was around 87%, 93%, and 94% higher than that of AM on days 3, 5, and 7, respectively.

The existence of a monocytic origin was evaluated to explore the macrophage populations presenting this difference in their proportion. As shown in Figure 1E, almost the entire population of IM was CD11b positive, while few AM presented expression on days 5 and 7. These results indicate that, throughout the inflammatory phase of BLM-induced pulmonary fibrosis, macrophage recruitment to the lung is intensified, and IM are the predominant macrophage population.

### 3.2. Inflammatory Phase of BLM-Induced Pulmonary Fibrosis Is Associated with a Differential Inflammatory Phenotype in Interstitial and Alveolar Lung Macrophages

Macrophages hold two distinct activation phenotypes (the classically designated M1- and M2-like phenotypes) that impact the fibrotic processes differently [9,22,23]. In this way, we next evaluated the profile of lung macrophages during the inflammatory phase of BLM-induced lung fibrosis, studying the cell surface expression of CD80 and MHCII as pro-inflammatory-associated molecules and CD206 as an anti-inflammatory by flow cytometry.

BLM-treated AM showed a higher proportion of CD80+ cells than IM (Figure 2A). The CD80 expression was more pronounced in both lung macrophage populations at day 3 after BLM administration (Figure 2A). Moreover, both IM and AM presented decreased CD80^+^ percentage throughout the disease progression. Interestingly, the levels of CD80 expression in IM, represented by mean fluorescence intensity (MFI), also decreased after day 3 post BLM (Figure 2A). These data suggest that the number of cells expressing CD80 decreases during the inflammatory phase in both macrophage populations.

IM from BLM-treated mice presented a statistically significant higher percentage of MHCII^+^ cells compared with AM on days 3 and 5 (Figure 2B). Furthermore, these differences were more greatly accentuated regarding the MFI levels of MHCII expression, inclusively, at 7 days after BLM administration (Figure 2B). However, the expression levels were unaltered within each lung macrophage population throughout the inflammatory phase, suggesting that none of the macrophage populations presents a higher number of MHCII^+^ cells over time.

Regarding the CD206-positive cell percentage, AM and IM from BLM-treated mice showed similar proportions at day 3. However, between days 5 and 7, the proportion of CD206^+^ AM was increasingly higher than that of CD206^+^ IM (Figure 2C). A similar result was observed in CD206 expression levels by MFI (Figure 2C). IM lost CD206 expression throughout the inflammatory phase, while AM presented an evident increase. These results indicate that CD206^+^ AM increased compared with CD206^+^ IM in the later inflammatory phase after BLM administration.

Our flow cytometry data hint that AM shift from the M1- to M2-like phenotype as CD80 expression levels decrease and CD206 expression levels increase throughout inflammation. On the other hand, these data do not suggest a clearly defined IM phenotype since CD80, and CD206 expression levels slightly decrease throughout the inflammatory phase of BLM-induced pulmonary fibrosis.

Further, a more detailed analysis of the activation profile of the lung macrophage populations at the peak of the inflammatory phase of BLM-induced pulmonary fibrosis was performed [7,8]. For that, the macrophages’ gene expression of pro-inflammatory (Il-1β, Tnf-α, Cxcl2, Ccl5, Ccl9, Ccl2), anti-inflammatory (Arg1, Chi3l3), and pro-fibrotic (Tgf-βi, Tgf-β1, Fn1, Mmp12, Ccl24) mediators were evaluated.

Treated IM displayed a significant decrease in relative expression levels of pro-inflammatory-related factors compared with both the vehicle (Il-1β, Tnf-α, Cxcl2, Ccl5) and with treated AM (Il-1β, Tnf-α, Cxcl2, Ccl9, Ccl2) (Figure 3A,D,E). Moreover, treated IM presented a significant increase in anti-inflammatory-related factors (Arg1, Chi3l3) compared with the vehicle control (Figure 3B). Regarding the relative expression levels of pro-fibrotic factors, no differences were found in treated IM compared with the vehicle. However, there was a decrease compared to AM (Figure 3C). These results suggest that IM are activated towards an anti-inflammatory phenotype.

Alveolar macrophages presented a significant increase in pro-inflammatory (Il-1β, Cxcl2, Ccl9, Ccl2), anti-inflammatory (Arg1, Chi3l3), and pro-fibrotic (Tgf-βi, Fn1, Mmp12, Ccl24) factors (Figure 3A, Figure 3B, and Figure 3C, respectively) compared with the vehicle. These results suggest that AM are in a transient phenotype while they still have a mixed phenotype. The protein expression of TNF-α and IL-1β was analyzed in treated IM and AM to validate gene expression levels. In fact, protein expression of both pro-inflammatory proteins was significantly increased in treated AM compared with treated IM (Figure 3D,E).

Our results show that BLM impacts the phenotype of IM and AM differently since the shift in activation state does not occur simultaneously. At this stage of BLM-induced pulmonary fibrosis, IM hold an M2-like phenotype, while AM are in a transient state from an M1- to an M2-like phenotype.

### 3.3. Activated Interstitial Macrophages Promote Proliferation and Fibroblast Differentiation

As the IM are the most representative lung macrophage population in the early phase of the mouse fibrosis model, next, their effects in fibrogenesis were investigated. Thus, IM were sorted from the lung at day 7 post BLM or post saline administration and subsequently cultured to collect the CM. The lung fibroblast cell line was stimulated with sorted, IM-derived CM, and proliferation and differentiation were examined using MTS assay and Ki-67 expression and expression of α-SMA and vimentin, respectively (Figure 4A). α-SMA is the most consistent marker for myofibroblast identification [24]. This cell type can also co-express α-SMA and vimentin; however, a single vimentin stain is reliable in distinguishing fibroblasts.

Fibroblasts stimulated with untreated and treated, sorted, IM-derived CM exhibited increased viability and proliferation compared with the control, as shown by MTS assay (Figure 4B). However, no differences were found in MLg to Ki-67 expression by immunofluorescence (Figure 4C,D).

After stimulation with sorted, IM-derived CM, single vimentin expression decreased and was mostly co-localized with α-SMA expression (Figure 4E,F) in both untreated and treated conditions. A significant increase in single α-SMA expression was observed when fibroblasts were stimulated with CM of IM sorted from BLM-treated animals (Figure 4E–H). These results suggest that activated IM modulate the proliferation of fibroblasts and promote their differentiation towards myofibroblasts, which are responsible for extracellular matrix production in pulmonary fibrosis.

## 4. Discussion

Pulmonary fibrosis is a common and severe pathological condition, and since there are no effective therapies to halt this condition, the discovery of new treatments is urgently needed. Therefore, a better understanding of macrophages’ signature and functionality in this context is critical to unraveling new therapeutic strategies.

Pulmonary fibrosis is a rapidly progressive condition characterized by the activation of fibroblasts and their differentiation into myofibroblasts. These cells are responsible for the excessive production and consequent accumulation of extracellular matrix [1,5,6]. Our study used the BLM-induced pulmonary fibrosis mouse model. BLM intratracheal administration is one of the best established and most widely used mouse models for studies that mimic human pulmonary fibrosis [25]. Intratracheal instillation of BLM is associated with the inflammation phase during the first week, reaching a peak at day 7, and a subsequent fibroproliferative phase leading to excessive collagen deposition [26]. Indeed, several studies demonstrated that this model shares cellular and molecular mechanisms important in the pathogenesis of idiopathic pulmonary fibrosis and other fibrotic interstitial lung human diseases [27,28,29]. We demonstrated in the early stage of this model that IM accumulate in the lung and become phenotypically activated in the disease progression. In vitro, this macrophage population leads to myofibroblast activation by paracrine mechanisms.

Our flow cytometry data in early BLM-induced pulmonary fibrosis showed that the proportion of IM increased throughout the disease and peaked at day 7. On the other hand, the proportion of AM decreased drastically and remained unaltered during these periods. These findings suggest the critical involvement of IM in the fibrogenesis process of the disease and a slightly residual role of the AM population. Moreover, our results show that the population of IM is positive for CD11b, while AM present a low percentage of positive staining. These results indicate that the increase in IM throughout the inflammatory phase of BLM-induced pulmonary fibrosis is due to their monocytic origin. This intensified recruitment and predominance of IM in the lung reinforces the potential involvement of this macrophage population rather than AM in the disease pathogenesis.

Macrophages present a colossal capacity to modulate their phenotype and function according to the tissue microenvironment. In this way, we investigated the respective activation state of AM and IM 7 days after BLM administration. Our results demonstrated that, until day 7, AM presented increasing expression levels of CD206 and decreasing expression levels of CD80. In addition, at the gene relative expression levels, several M1-, M2-like, and pro-fibrotic factors were simultaneously elevated. Other reports presented the same tendency, such as Misharin et al. [20], who suggested that, during the early stages of BLM-induced pulmonary fibrosis mouse model, an M1-like response occurs. Instead of its cessation, the M2-like response begins simultaneously in the following stages, and then the M1-like response ceases at later fibrotic stages. This study showed that, in the early stages of BLM-induced pulmonary fibrosis (5 days), AM express higher levels of CD206 and CD80 than the vehicle control, while, in later stages (21 days), AM only express higher levels of CD206 than the vehicle control. Zhang et al. [30] described that, on day 14 after BLM administration, AM had higher relative expression levels of Arg1 and lower expression levels of Tnf-α than vehicle control. Furthermore, Zhu et al. [31] stated that, on day 17 after BLM administration, AM expressed higher CD206 levels than the control. These data indicate that, at early stages of BLM-induced lung fibrosis, AM are transient state from an M1-like to an M2-like phenotype.

Regarding IM, our data showed that IM have slightly decreased levels of CD206 and CD80 expression during the inflammatory phase. Additionally, at the gene relative expression levels, several pro-inflammatory factors were reduced, and anti-inflammatory factors were elevated compared with the vehicle control, which is in accordance with previous studies on day 10 after BLM administration. Misharin et al. [20] showed that, on day 5 after BLM administration, IM have increased CD80 expression and lower CD206 expression compared with the vehicle control, and on day 21, no differences were found in the expression of these markers. Ucero et al. [32] and Zhang et al. [33] described that, at day 10 after BLM administration, IM have higher relative expression levels of CD206 and Arg1 compared with the vehicle control, suggesting that, at this phase, IM are activated towards an M2-like phenotype. In line with this, our findings revealed that, at day 7, IM were activated towards an M2-like phenotype.

Since IM accumulate in BLM-treated mice’s lungs, M2-like macrophages are associated with aberrant tissue healing and consequent fibrosis through the secretion of anti-inflammatory and pro-fibrotic factors [7,22,23,34], we further investigated the role of IM in fibrogenesis using in vitro experiments. We showed that CM from sorted IM of BLM-treated mice promoted fibroblasts proliferation and differentiation into myofibroblasts. Our findings comply with other studies. For instance, Meziani et al. [18] used sorted IM (F4/80^+^ Gr1^−^) from radiation-induced lung fibrosis and activated them with interferon-gamma to induce the M1-like phenotype or IL-13/IL-4 to induce the M2 phenotype. After being co-cultured with M2-activated IM, normal fibroblasts overproduce extracellular matrix components and increase their α-SMA expression. On the other hand, co-culturing with M1-activated macrophages does not affect either fibroblasts phenotype or function. Hou et al. [35] performed co-cultures of lung-resident mesenchymal stem cells (LR-MSCs), with M1 or M2 macrophages polarized, using a mouse macrophage cell line, RAW264.7. This study showed that co-culturing with M2 macrophages induces LR-MSC differentiation, evidenced by the increased α-SMA expression, while M1 macrophages do not alter LR-MSCs. Both studies suggested that M2-like macrophages, rather than M1-like macrophages, induce differentiation into myofibroblasts, because we observed that IM were activated towards an M2-like phenotype at this point in the disease model. Our results showed that the IM population acts as a fibrogenesis promotor in the early stages of BLM-induced pulmonary fibrosis. In contrast, they were able to promote fibroblast activation in vitro, probably through their activated anti-inflammatory and downregulated pro-inflammatory phenotype.

## 5. Conclusions

Our study revealed the dynamic profile of AM and IM in their proportions during the early stages of a BLM-induced pulmonary fibrosis mouse model, with IM being the more numerous lung macrophage population. Additionally, our data confirmed the AM/IM activation phenotype in this model. Importantly, we exposed the influence of IM at the early stages of BLM-induced pulmonary fibrosis on promoting fibrotic processes by paracrine mechanisms.

## Figures and Tables

**Figure 1 cells-12-00402-f001:**
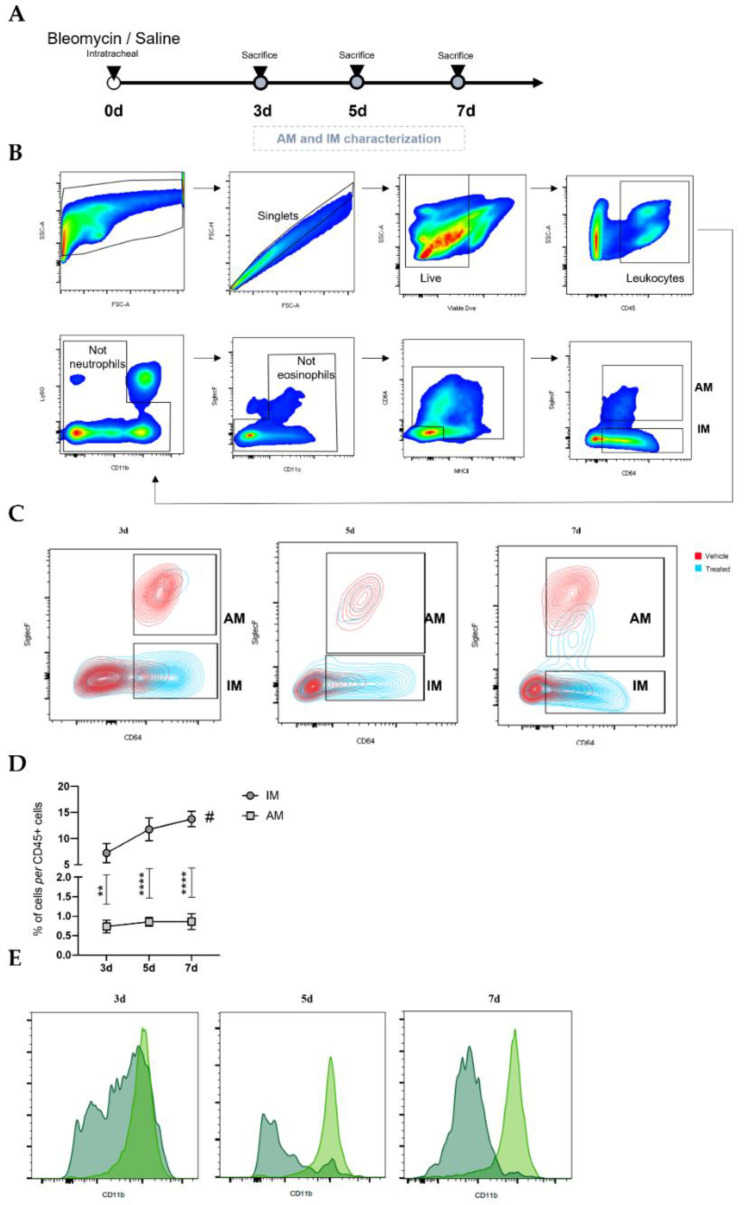
Macrophage populations in the lung during the inflammatory phase of BLM-induced pulmonary fibrosis mouse model. (**A**) Intratracheal injection of BLM or saline was performed on day 0. Animals were sacrificed on days 3, 5, and 7 to quantify macrophage populations. (**B**) Gating strategy for the identification of AM and IM in lung samples by flow cytometry. (**C**) Representative plots of proportions of AM and IM in untreated (red) and BLM-treated (blue) lungs at days 3, 5, and 7. (**D**) Proportion of IM and AM in BLM-treated animals at days 3, 5, and 7. The percentage of cells in live CD45+ is plotted. (**E**) Profile of CD11b expression in IM (light green) and AM (dark green) on BLM-induced PF mice at days 3, 5, and 7. Data are shown as the mean ± SEM of one representative experiment of three independent experiments. ** *p* < 0.01, **** *p* < 0.0001: treated IM vs. treated AM. # *p* < 0.05: different time points. IM: interstitial macrophages; AM: alveolar macrophages; d: days post intratracheal BLM administration.

**Figure 2 cells-12-00402-f002:**
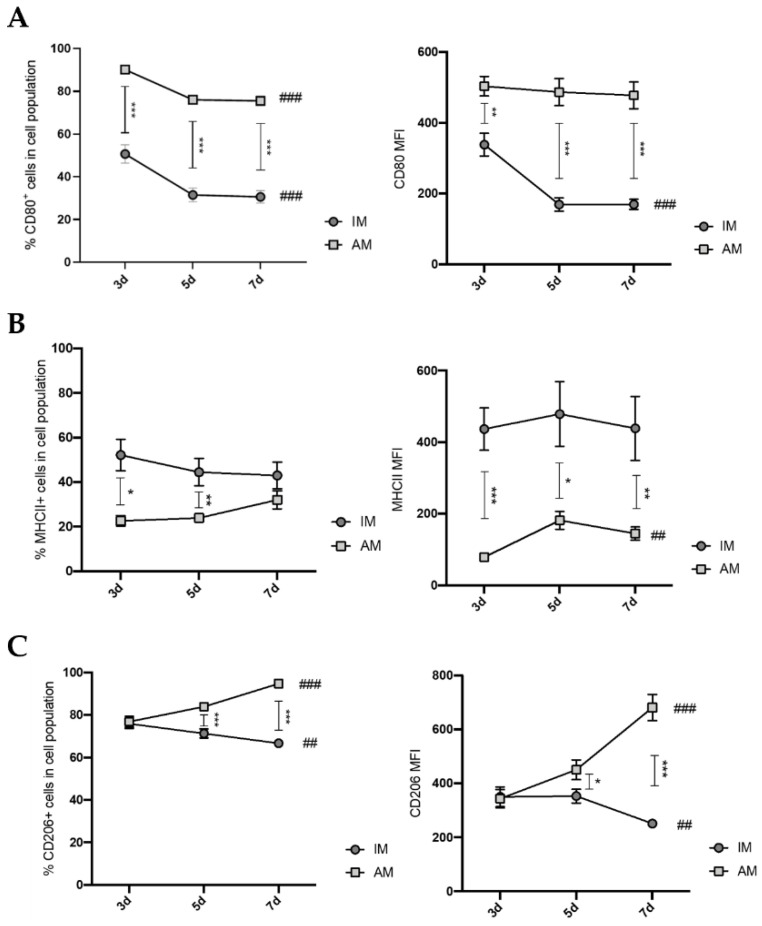
Macrophage populations phenotype in the lung during the inflammatory phase of BLM-induced pulmonary fibrosis mouse model. Expression profile of (A) CD80, (B) MHCII, and (C) CD206 in interstitial- and alveolar-macrophage-treated lungs at days 3, 5, and 7 after BLM administration. The percentage of cells was gated in live CD45 cells. The mean fluorescence intensity (MFI) in the respective macrophage population is plotted in the second graph. The data are represented as the mean ± SEM. * *p* < 0.05, ** *p* < 0.01, *** *p* < 0.001: treated IM vs. treated AM. ## *p* < 0.01, ### *p* < 0.001: different time points.

**Figure 3 cells-12-00402-f003:**
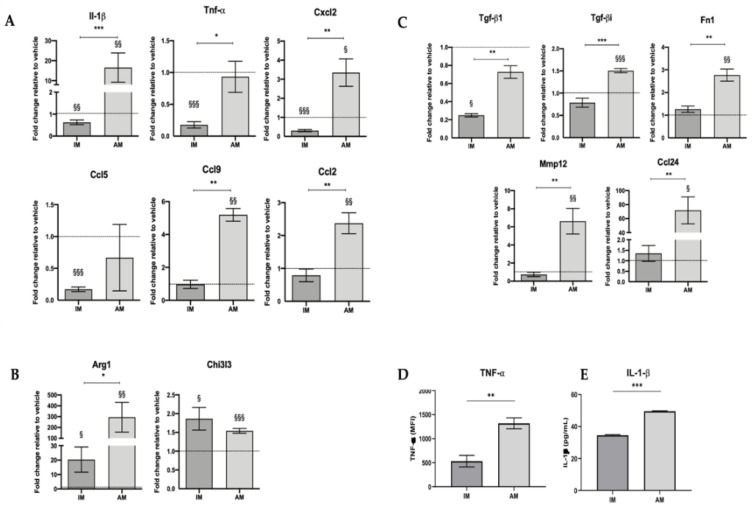
IM are activated towards an anti-inflammatory phenotype, and AM are in a transient activation state at day 7 after BLM-induced pulmonary fibrosis. Gene expression profile of interstitial and alveolar macrophages sorted from BLM and saline-injected lungs at day 7 of pro-inflammatory (**A**), anti-inflammatory (**B**), and pro-fibrotic factors (**C**). The fold change of gene relative expression of the respective group vehicle was set to 1. TNF-α protein expression was obtained by the mean fluorescence intensity (MFI) using the flow cytometry gating strategy in Figure 1B (**D**), and expression of IL-1β was obtained by ELISA assay of IM-derived and AM-derived CM after 24 h culture (**E**). Data are represented as the mean ± SEM. * *p* < 0.05, ** *p* < 0.01, *** *p* < 0.001: treated IM vs. treated AM. § *p* < 0.05, §§ *p* < 0.01, §§§ *p* < 0.001: treated vs. vehicle.

**Figure 4 cells-12-00402-f004:**
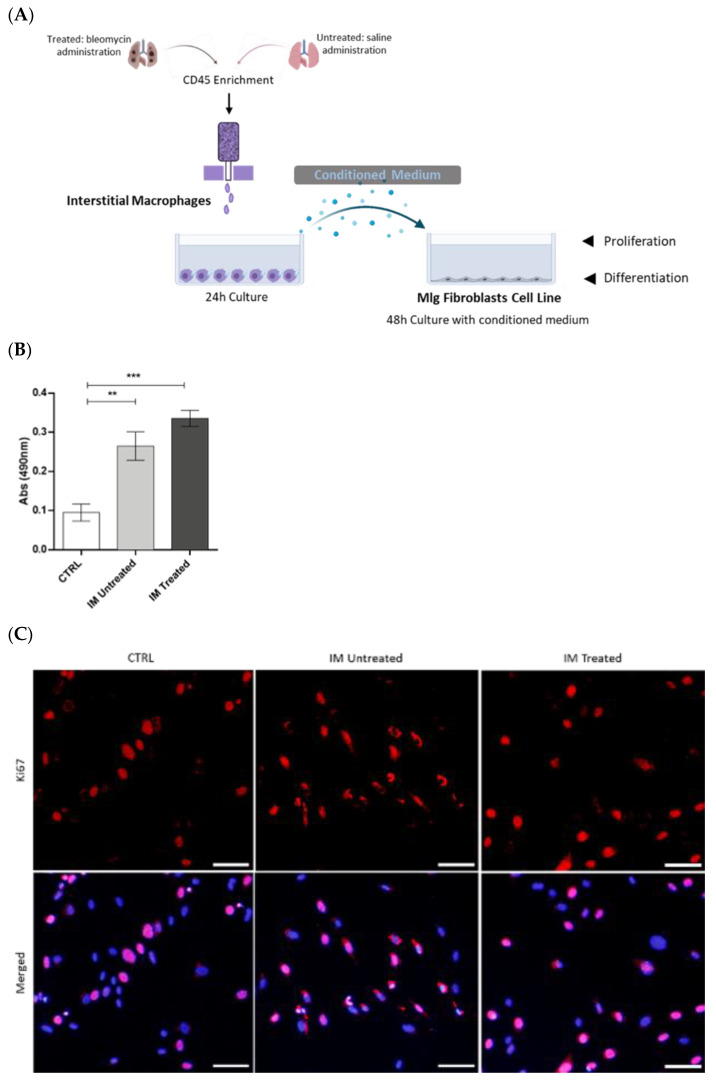
Conditioned medium from BLM-sorted IM induce fibroblasts proliferation and differentiation into myofibroblasts. (A) Experimental design. After CD45 enrichment, IM were sorted from dissociated lung samples on day 7 after saline (IM untreated) or BLM (IM treated) administration and cultured for 24 h to collect the CM. MLg, a fibroblast cell line, was cultured with control medium, conditioned medium from untreated IM, or treated IM. After 48 h, the (B) metabolic activity was measured by MTS assay, (C) cell proliferation by Ki-67 staining, and (D) respective quantification was performed. Differentiation state was analyzed by (E) immunofluorescence with vimentin (red) and α-SMA (green), nuclei are stained with DAPI (blue), and (F) respective quantification, also by (**G**) Western blot, and (**H**) quantification of relative expression. Scale bar = 200 µm. Original magnification, 100×. Data are shown as the mean ± SEM of one representative experiment of three independent experiments. ** *p* < 0.01, *** *p* < 0.001: treated IM vs. treated AM.

## Data Availability

The authors will freely release all data supporting the published paper upon direct request to the corresponding author.

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
