# Peer review of "Interstitial Macrophages Lead Early Stages of Bleomycin-Induced Lung Fibrosis and Induce Fibroblasts Activation"

_cells, 2023, doi:10.3390/cells12030402_

Round 1
Reviewer 1 Report
In this manuscript, the authors proposed that interstitial macrophages are the highest macrophage population at early stages of experimental pulmonary fibrosis and secreted mediators able to activate fibroblasts. The topic and content of this work are relevant to the scope and readership of the journal, and studies are generally well designed and executed. The manuscript could be considered for publication if the following concerns (listed below) are addressed.
1. Please include the number of mice used in this study.
2. The space between “mm” and “3” is missing in line 102.
3. In figure 4, immunostaining showed that sorted IM-derived CM could promote fibroblast to myofibroblast differentiation. It is better adopting western blot or RT-PCR to confirmed these results.
Author Response
We thank the reviewer for reconsidering the revision of our manuscript for publication in Cells journal. We are certain to have improved our previous version of the manuscript with the points exposed by the reviewer.
We are now submitting a revised version in which all the points raised by the two reviewers have been addressed.

Reviewer 2 Report
The manuscript “Interstitial Macrophages Lead Early Stages of Bleomycin-In- 2 induced Lung Fibrosis and Induce Fibroblasts Activation” by Sofia et al., was well written. The authors used both In vivo and In vitro models to prove their hypothesis. Let me make some comments.
Major Comments:
1. Authors should justify why they selected the 7-day trial for studies? What are experiments conducted to know that IPF is established in the tissue? The first result should be regarding the fibrosis induction.
2. Authors have claimed the mechanism only based on qPCR which is hard to believe. Only based on one experiment authors have written the mechanism, I request them to provide protein expression data for at least two main markers like TNF-alpha IL-1β and TGF-beta (in figures C TGF-beta is labelled twice. And authors should mention gene expression data rather than transcript)
3. Authors should cite the reference for the CM based experiments on fibroblast.
4. In the results 2. authors claims transient phase of AM to IM. Arginase I is the perfect marker to show this phase transition (western/qPCR experiment)
5. Vimentin/smooth muscle markers used are the intracellular marker, if you show the secreted marker fibronectin & laminin markers may be useful in defining the fibroblast differentiation phase to myofibroblast.
6. Why there is no discussion of MMPs which are secreted by AM, which are important in fibrogenesis? Why exclusively used MMP-12 as marker?
Minor comments
1. The symbols used micro should be µ not u.
.
Author Response

(The authors gave the same response as above.)

Round 2
Reviewer 2 Report
The manuscript “Interstitial Macrophages Lead Early Stages of Bleomycin-In- 2 induced Lung Fibrosis and Induce Fibroblasts Activation” by Sofia et al., is improved but needs to incorporate few more points. I happy with the revision and please answer a few more points.
Comments
1. Authors claims that so many literature are published, is there any model established and published from there lab or group? If so please cite the reference or give two results to show the fibrosis at 7 day.
2. “Vimentin/smooth muscle markers used are the intracellular marker, if you show the secreted marker, …..the answer for this comment is not justifiable. If they provide sufficient justification, it will be good.
Thanks
Author Response
The responses to the minor comments are found in the PDF file.
Best regards
